# Classification and Distribution of the Dayside Ion Upflows Associated with Auroral Particle Precipitation

Yao Yu [1,2], Ze-Jun Hu [2,3,*], Hong-Tao Cai [1] and Yi-Sheng Zhang [4]

1   Electronic Information School, Wuhan University, Wuhan 430072, China
2   MNR Key Laboratory for Polar Science, Polar Research Institute of China, Shanghai 200136, China
3   Ocean College, Zhejiang University, Zhoushan 316021, China
4   Beijing Institute of Applied Meteorology, Beijing 100029, China
*   Correspondence: huzejun@pric.org.cn

**Abstract:** Two important phenomena of the solar wind–magnetosphere–ionosphere coupling are auroral particle precipitation and the formation of ions flowing upward from the ionosphere. They have opposite transport directions of energy and substance. Based on the observations of particle precipitation and ion drift from the DMSP F13 satellite in January and July 2005, the ionospheric ion upflows in dayside auroral oval (0600–1800 MLT) can be divided into five types according to the velocity of ion upflows and the spectrum characteristics of auroral particle precipitation, and the distribution for different types of ion upflows is studied. The results show that the ion upflows mainly occur in the geomagnetic latitude (MLAT) range of 70–80°. The main magnetospheric source region of ion upflows (type A and D) caused by the accelerated electron (mainly the soft electron) corresponds to Low Latitude Boundary Layer (LLBL) and Cusp, and ion upflows of type B and C (related to the process of ambipolar diffusion caused by electron acceleration) mainly occur in LLBL and Boundary Plasma Sheet (BPS), while ion upflows of type E without electron acceleration mainly occur in the central plasma sheet (CPS). The dawn–dusk asymmetry is obvious in the winter season, with the ion upflows mainly occurring on the dawn/dusk side ionosphere. However, the ion upflows in summer mainly occur at the magnetic noon, with a symmetric distribution centered at the magnetic noon. The occurrence of ion upflow in winter is significantly higher than that in summer, and it is significantly enhanced during the period of moderate geomagnetic activity. The upward region expands to the lower latitude when the geomagnetic activity is enhanced. The effect of interplanetary magnetic field (IMF) components has also been studied in this paper. When IMF $Bx$ is negative, the upflow occurrence increases in the region of 1500–1800 MLT and 0600–0900 MLT, with the MLAT range below 70°. The direction of IMF $By$ may lead to the high-incidence area reverse at the prenoon or postnoon region. The occurrence of ion upflows with the MLAT range below 75° increases significantly when IMF is southward. Type A ion upflow has the highest velocity of ion upflows, followed by type E, and type D has the lowest. The average velocity of ion upflows in winter is significantly higher than that in summer.

**Keywords:** particle precipitation; ion upflow; geomagnetic activity; interplanetary magnetic field





## 1. Introduction

Through the interaction of solar wind, interplanetary magnetic field and Earth's magnetosphere, part of the energy, mass and momentum carried by solar wind enters the magnetosphere through various dynamic processes. These dynamic processes mainly occur in various boundary layers and magnetotails and are mapped to the polar ionosphere via magnetic field lines. As the most intuitive ionospheric trace characterizing various magnetospheric dynamical processes, aurora is extremely important for the study of space weather and the coupling between solar wind and the magnetosphere. Since the ionospheric projection of each magnetospheric boundary layer on the dayside is located on the auroral

oval [1,2], the morphology, spectrum, intensity, motion and other characteristics of dayside aurora are closely related to the various dynamic processes of magnetospheric boundary layer on dayside. However, how each dynamic process accelerates the formation of dayside aurora is unclear. Newell et al. [3] divided the discrete auroral electron precipitation into two types, namely, a monoenergetic acceleration event and a broadband acceleration event, according to the electron spectrum characteristics of precipitating particles. In addition, these two types of precipitation correspond to the process of quasi-static acceleration and dispersion Alfvén acceleration, respectively.

The ion upflows in the polar ionosphere are another important phenomenon in the solar–earth energy coupling system. The $O^+$ and $NO^+$ in lower atmosphere/ionosphere are transported to the magnetosphere through the upflow of ions along the magnetic field lines, affecting the characteristics of magnetosphere plasma and related dynamic processes. Ion accelerated to escape velocity, namely ion outflows, is an important path of material exchange between earth and interplanetary space. It has been proved that there are a variety of ion outflows and ion energy acquisition processes in the polar ionosphere. The components of ion outflow include low-energy ions ($H^+$, $He^+$) and high-energy ions ($O^+$, $NO^+$, $O^{2+}$), which play an important role in the coupling of the ionosphere and the magnetosphere [4,5]. Since single-charged oxygen ions do not exist in solar wind, $O^+$ in magnetospheric plasma mainly come from the ionization of oxygen atoms in the ionospheric F region [6–8].

In terms of energy transport, the aurora particle precipitation is opposite to the energy transport of ion upflows. However, there is a close relationship between particle precipitation and ion upflow, especially dayside auroral particle precipitation. One of the main driving factors of dayside ion upflows is the precipitation of soft electrons in the polar gap region. Soft electrons (<500 eV) enter the polar gap region, ionize neutral particles and generate heated electrons, which move upward with ions through the bipolar electric field [9]. Statistical studies show that the maximum ion flux occurs in the ~78° ILAT and 0900–1500 MLT sectors of the auroral oval on dayside [10], and the incidence of the ion upflows peaks at 0800 MLT and 1300 MLT [11]. The 0900–1500 MLT sector is the soft electron precipitation region, with strong emission at 630.0 nm of dayside red coronal aurora [12,13], while 0800 MLT and 1300 MLT correspond to the prenoon "warm spot" and postnoon "hot spot" regions on dayside auroral oval, respectively. The auroral morphology of these two areas is dominated by green auroral arcs and complex hot spot aurora [2,12–14]. Ground-based optical observations show that there is a one-to-one relationship between poleward moving auroral forms (PMAFs) and ion upflows in the cusp region, suggesting that individual events of soft electron precipitation trigger corresponding ion upflows [15].

The type difference of aurora corresponds to the energy spectrum characteristics of different particle precipitation; that is, the average energy, energy flux and energy level range of the particles are different. The difference in these parameters may lead to different parameter characteristics in ion upflows. The relationship between the particle precipitation of so many different types and the ion upflows is not clear. Therefore, the systematic study of the relationship has important theoretical significance and practical value for understanding the coupling of magnetosphere and ionosphere.

In this paper, data from the DMSP F13 satellite are used to study the ion upflow and particle precipitation of the northern hemisphere, mainly to investigate the temporal–spatial distribution of ion upflows corresponding to different precipitation electron spectrum characteristics, and preliminarily discuss the physical mechanism of such distribution differences.

## 2. Data and Analysis

The satellites of the Defense Meteorological Satellite Program (DMSP) are a series of polar-orbiting satellites created for use by the U.S. Air Force to monitor the state of the environment in the near-Earth space on a "near real-time" basis. The first DMSP satellite

was flown in the early 1960s, taking 101 min to orbit the earth. The orbit altitude is about 835–850 km, and the inclination Angle is about 96°. Special Sensor for Particle Flux (SSJ/4) has 19 energy channels (34, 49, 71, 101, 150, 218, 320, 460, 670, 960 eV and 1.4, 2.1, 3.0, 4.4, 6.5, 9.5, 14.0, 20.5, 29.5 keV, respectively) and can measure the energy fluxes of precipitating electrons and ions in the energy range of 30 eV–30 keV, with data recorded every one second [16–19]. SSIES (Special Sensors for Ions, Electrons, and Scintillations) measure three components of the plasma flow velocity, plasma density and temperature of ion and electron [20]. The orbit of DMSP F13 satellite mainly covers the dawn and dusk side of the northern hemisphere, so the distribution of ion upflows on both sides can be studied.

Newell [21–23] summarized the energy spectrum characteristics of particle precipitating in different source regions. According to the energy spectrum data of DMSP SSJ/4 particle precipitating, the source region of the magnetosphere corresponding to the particle precipitation can be judged as the follows: (1) Cusp region: the average ion energy is 300–3000 eV, the average electron energy is less than 220 eV, and the total energy flux of ion and electron (eV/cm$^2$·s·str) is less than $10^{10}$ and $6 \times 10^{10}$, respectively [21]; (2) Low Latitude Boundary Layer (LLBL): the average ion energy is 3000–6000 eV, the average electron energy is 220–600 eV, and the total energy flux (eV/cm$^2$·s·str) of ions and electrons is less than $10^{10}$ and $6 \times 10^{10}$, respectively [21]; (3) Plasma mantle: the ion energy is less than 100 eV, and the density is $10^{-2}$–$10^{-1}$/cm$^3$. The average energy and total flux of particles decrease with the increase in the geomagnetic latitude [22]; (4) Boundary Plasma Sheet (BPS): the electron energy is less than 1 keV, and the total energy flux of ions and electrons (eV/cm$^2$·s·str) is more than $10^{10}$ [23]; (5) Central Plasma Sheet (CPS): the electron energy is greater than 1 keV, and the total energy flux of ions and electrons (eV/cm$^2$·s·str) is greater than $10^{10}$ [23].

Ion velocity or flux can be used as the criterion for ion upflows. When the velocity or flux is much higher than the normal value, it can be judged as ion upflow [24]. Without additional acceleration, the upward drift velocity of ions at 800 km is generally lower than 200 m/s [25]. When the upward drift velocity is greater than 200 m/s, an upward event is considered, with the possible offset in the baseline of the measured data of SSIES. In order to reduce the interference of noisy signals (a single data point has a high velocity, while nearby data points have no upflow characteristics), and considering that the resolution of velocity and ion density in the measured data is 1 s and 4 s, respectively, the ion upflows studied in this paper contain at least four consecutive data points with a velocity no less than 200 m/s.

At high latitudes, the vertical velocity $V_z$ approximates the ion velocity $V_b$ along the magnetic field [26]. However, since most typical ion upflows occur in the lower latitudes (60°–70° MLAT), the approximate replacement of $V_b$ with $V_z$ will introduce some error. Using the International Geomagnetic Reference Field (IGRF-11) model, we can calculate the components of the Earth's magnetic field: $Bx_1$ (northward component of the magnetic field), $By_1$ (eastward component of the magnetic field), $Bz_1$ (downward component of the magnetic field). The velocity $V_b$ along the magnetic field can then be calculated by projecting the vertical velocity $V_z$ onto the magnetic field.

The electron precipitation with accelerating characteristics is defined as follows: in 19 energy channels at a point in time, as long as the flux of any energy level is greater than $10^8$ eV·cm$^{-2}$·s$^{-1}$·str$^{-1}$·eV$^{-1}$, the electron spectrum is considered to have accelerating characteristics [3]. When the flux of each energy level in the electron accelerating structure is similar and the structure is geometrically connected in the energy spectrum, the electron accelerating characteristic structure is determined to be continuous. In DMSP data, there are different corresponding characteristics between the velocity of ion upflows and the structure of electron precipitation. According to the different corresponding relations, the ion upflows can be further classified as follows: (1) Type A: The upflow velocity has an obvious peak interval, and the whole peak interval should have obvious single and continuous electron acceleration structure; (2) Type B: The upflow velocity has an obvious peak interval, corresponding to multiple and scattered electron acceleration structures;

(3) Type C: The upflow velocity has an obvious peak interval, but only the ascending or descending segment corresponds to the structure of electron acceleration; (4) Type D: the upflow velocity corresponds to the structure of electron acceleration, but the velocity has no obvious change trend or complete peak interval; (5) Type E: the upflow velocity does not correspond to the structure of electron acceleration.

Figure 1 shows different types of ion upflows. The upper is the upflow velocity, the lower is the electron energy spectrum, the horizontal blue dashed line is the threshold value of 200 m/s and the red box corresponds to the occurrence area of ion upflows. The upflow velocities in 'a' during 08:20:41–08:20:49 UT are greater than 200 m/s, and the corresponding electron spectrum show continuous electron acceleration feature. According to the determination criteria, this event was judged as a Type A ion upflows. Similarly, the red box in 'b' corresponds to the time during 16:44:33–16:44:38 UT, which is Type A ion upflow, while 'c' and 'd' are Type B ion upflows. The electron spectra show multiple scattered electron-accelerating characteristics. 'e' and 'f' are Type C ion upflows, and the electron spectra in the red boxes show a continuous electron acceleration feature, but only corresponding to the regions where the upward velocity decreases or increases. 'g' and 'h' are Type D ion upflows. The upflow velocity in the red box of 'g' shows no obvious variation trend, and the electron spectrum shows continuous electron accelerating characteristics. The velocity in 'h' around 11:36:32 UT is lower than 200 m/s, leading to an increasing trend without a complete peak interval. 'i' is Type E ion upflow, and the electron spectrum in the red box does not have acceleration characteristics.

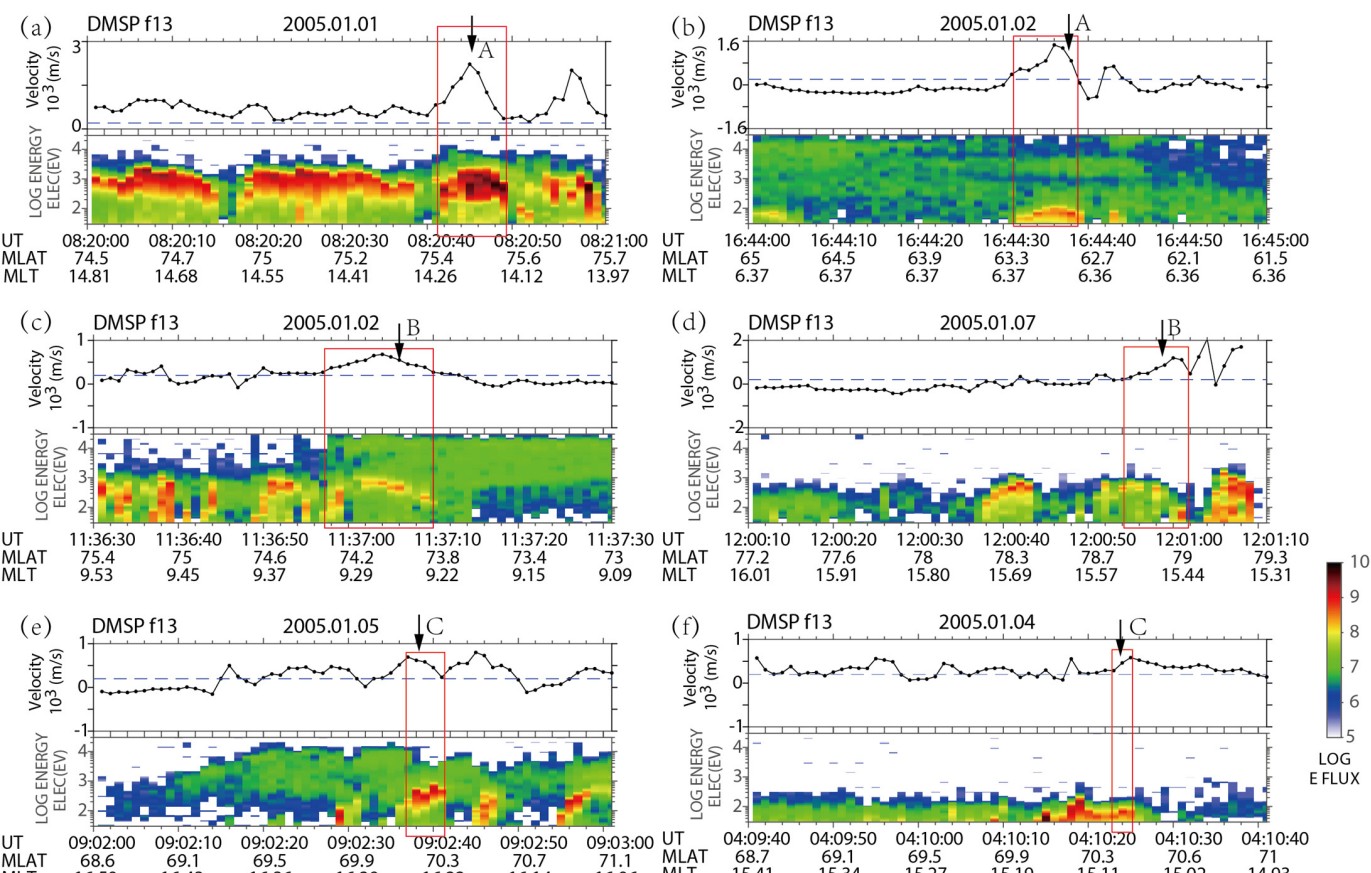

**Figure 1.** *Cont.*

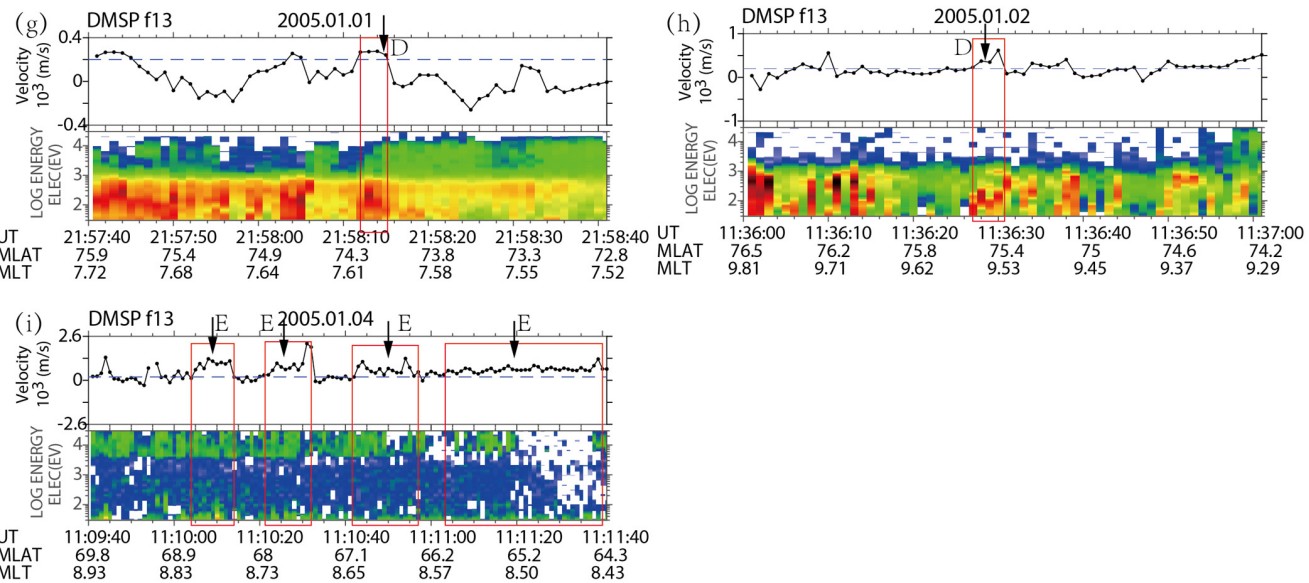

**Figure 1.** Corresponding relationship between the vertical ion drift velocity (top panel) and electron acceleration (bottom panel) for different types of ion upflows. The horizontal blue dashed line is the threshold value of 200 m/s. The red box indicates the interval of typical ion upflows, and the black arrow and the letters (A–E) next to it indicate the type of ion upflows. (**a**,**b**) Type A ion upflows, (**c**,**d**) type B ion upflows, (**e**,**f**) type C ion upflows, (**g**,**h**) type D ion upflows, (**i**) type E ion upflows.

## 3. Statistical Results

### 3.1. Characteristics of Magnetosphere Source Region

Based on the data from the DMSP F13 satellite, a total of 15,198 ion upflows occurring in the range of 60~90° MLAT in the Northern Hemisphere during January and July 2005 are statistically studied. Among them, 5824 are type A, 637 are type B, 523 are type C, 1865 are type D and 6349 are type E. The specific values are shown in Table 1. Figure 2 shows the number distribution of different types of ion upflows corresponding to different magnetospheric source regions.

**Table 1.** Statistics of ion upflows in the geomagnetic latitude range of 70–80° on the dayside region of northern hemisphere in January and July 2005. From left to right, the source region of the magnetosphere is, successively, polar rain (prn), plasma mantle (mantle), polar gap (Cusp), low-latitude boundary layer (LLBL), plasma sheet boundary layer (BPS) and central plasma sheet (CPS), the same below, while the others are the events with no clear magnetospheric source region, and sum is the total number.

|     | Prn | Mantle | Cusp | LLBL | BPS | CPS | Others | Sum |
|-----|-----|--------|------|------|-----|-----|--------|-----|
| A   | 16  | 190    | 370  | 1343 | 2208 | 138 | 1159 | 5824 |
| B   | 3   | 10     | 4    | 173  | 239 | 17  | 191  | 637 |
| C   | 1   | 4      | 2    | 131  | 208 | 17  | 160  | 523 |
| D   | 6   | 79     | 112  | 457  | 696 | 48  | 467  | 1865 |
| E   | 19  | 131    | 3    | 666  | 1072 | 1246 | 3212 | 6349 |
| sum | 45  | 414    | 491  | 2770 | 4423 | 1466 | 5589 | 15,198 |

It can be seen from Table 1 and Figure 2 that all kinds of ion upflows with electron acceleration characteristics mainly appeared in regions of BPS, LLBL, Cusp and plasma mantle on dayside, while type E without electron acceleration mainly appeared in CPS, BPS and LLBL on dayside. Cusp is the main region for type A and D. Type B and C mainly appear in LLBL and BPS, and CPS is the main region for type E. In addition, there were clear seasonal differences in the number of different types of ion upflows. For type C and

E, the number in January (winter) (486, 5884, respectively) is much higher than that in July (summer) (37, 465, respectively), which is about 10 times greater than that in each magnetosphere source region. For type B, the number in January (532) was five times higher than that in July (105). For type A and D, the number in January (3654 and 1086, respectively) was slightly higher than that in July (2170 and 779, respectively): less than two times.

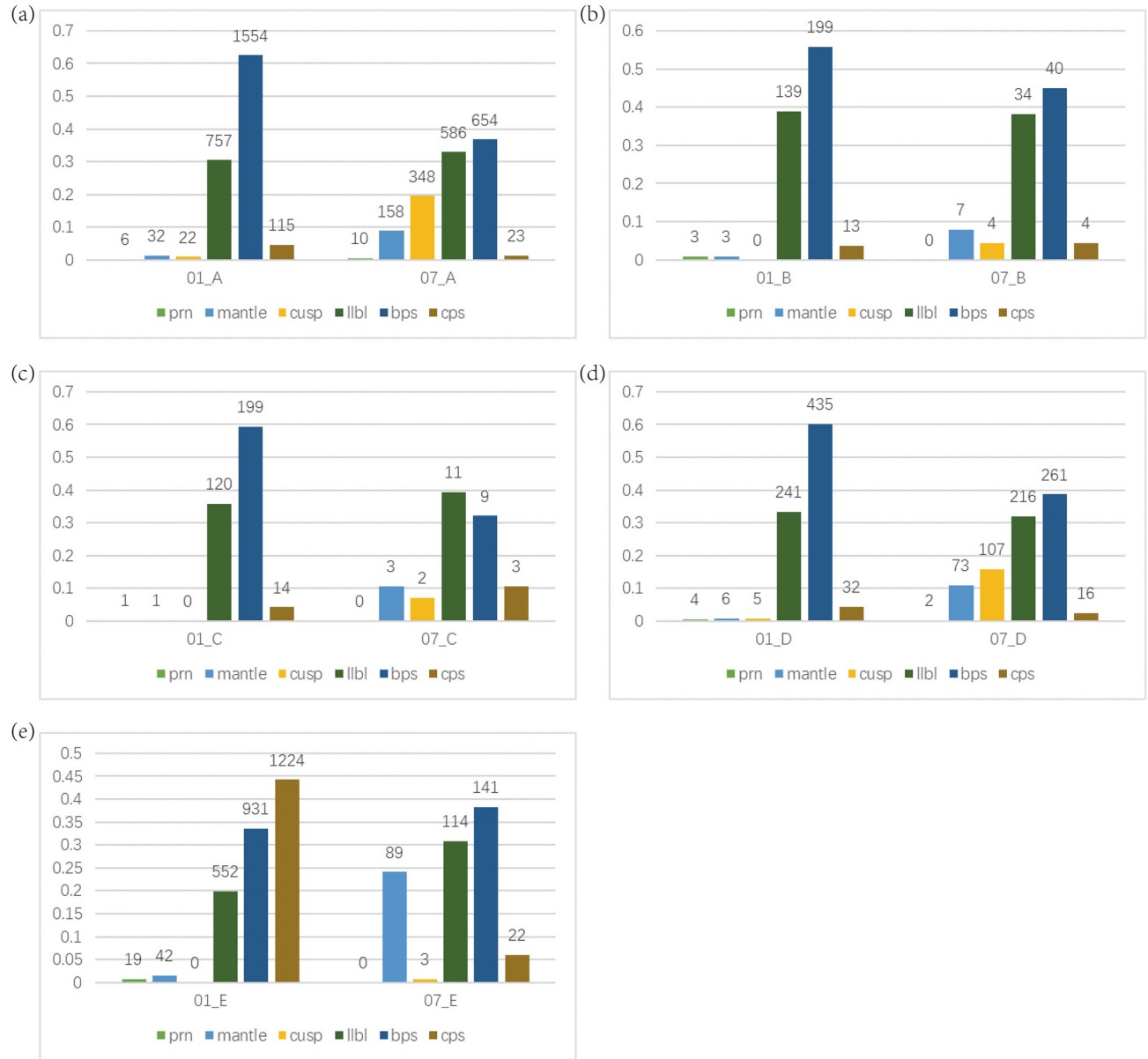

**Figure 2.** The distribution of ion upflow events corresponding to different magnetospheric source regions. (**a**–**e**) indicates type A–E ion upflows, respectively. (**left**) Statistics for January; (**right**) statistics for July.

## 3.2. Characteristics of Incidence

The Altitude Adjusted Corrected Geomagnetic (AACGM) coordinate plane is meshed to study the distribution characteristics of ion upflows. MLAT is divided into 12 groups, ranging from 60° to 90°, with an interval of 2.5°, and MLT is divided into 24 groups, ranging from 6 to 18 h, with an interval of 0.5 h. When the orbit of DMSP satellite passes through a grid, the number of traverses of the grid is increased by one. For the events that span two

or more grids, the number of traverses is added by 1 for each grid through which the event passes. We define the average orbital incidence *F* as:

$$F = N_{up}/N_{dmsp},$$ (1)

$N_{up}$ is the number of ion upflows observed by DMSP in the selected grid, and $N_{dmsp}$ is the number of orbital passes of DMSP satellite in the grid. To ensure the validity of statistical results, when $N_{dmsp}$ in a grid is less than 20 times, $N_{up}$ in the grid does not participate in statistics.

### 3.2.1. The Distribution and Occurrence for Different Types of Ion Upflows

Figure 3 shows the temporal–spatial distribution of the number of events and the average orbital incidences of all kinds of ion upflows in January/July. The gray area is the coverage of satellite orbit, and there are no data in the blank, the same as below. In the range of 70–80° MLAT, the incidences of type A (January/July) are 44.22 and 25.79, that of type B are 7.62 and 1.51, that of type C are 6.55 and 0.53, that of type D are 13.43 and 9.58 and that of type E are 59.39 and 6.17, respectively. In the sum of the occurrence rates of other latitudes (January/July), type A are 10.70 and 6.43, type B are 1.35 and 0.43, type C are 1.16 and 0.12, type D are 3.18 and 2.75 and type E are 51.69 and 1.97. This indicates that the occurrence rate of type A–D is the highest in the range of 70–80° MLAT, which is 3–6 times the total occurrence rate of other latitude ranges, while type E has high incidence in the region above 65° MLAT.

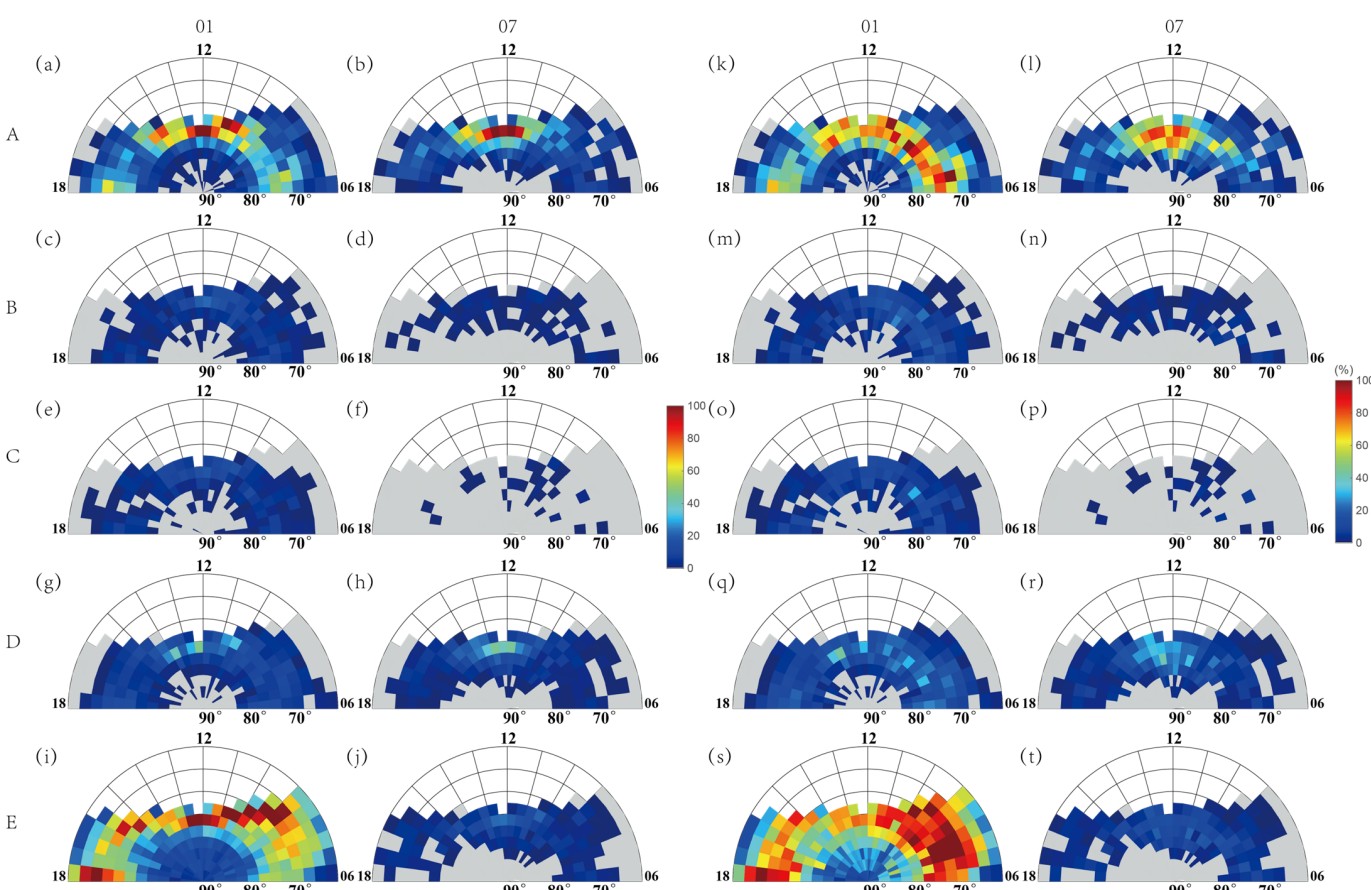

**Figure 3.** The distribution and occurrence for different types of ion upflows in January (left panel)/July (right panel) in MLT/MLAT coordinate. From top to bottom are ion upflows of type A to E, respectively. (**a**–**j**) Total events for each month and type, (**k**–**t**) average orbital occurrence in different month and type.

In January, all kinds of ion upflows are mainly distributed at dawn and dusk, and the incidence of dawn is significantly higher than that of dusk, showing obvious "dawn–dusk asymmetry". The incidence peak of type A, B and C is near 0900 MLT, and that of type D is near 1400 MLT. The incidence of type E peaks near 0600–0900 MLT and 1800 MLT. The ion upflows in July are mainly distributed around magnetic noon, with a symmetric distribution centered at the magnetic noon.

### 3.2.2. Temporal–Spatial Distribution of Different Types of Ion Upflows under Different Geomagnetic Activities

Figure 4 shows the distribution of the average orbital incidences of various ion upflows in AACGM coordinate plane under different geomagnetic activity conditions. $Kp < 2$ refers to quiet magnetic activities, $2 \leq Kp \leq 4$ refers to moderate magnetic activities, and $Kp > 4$ refers to disturbed magnetic activities.

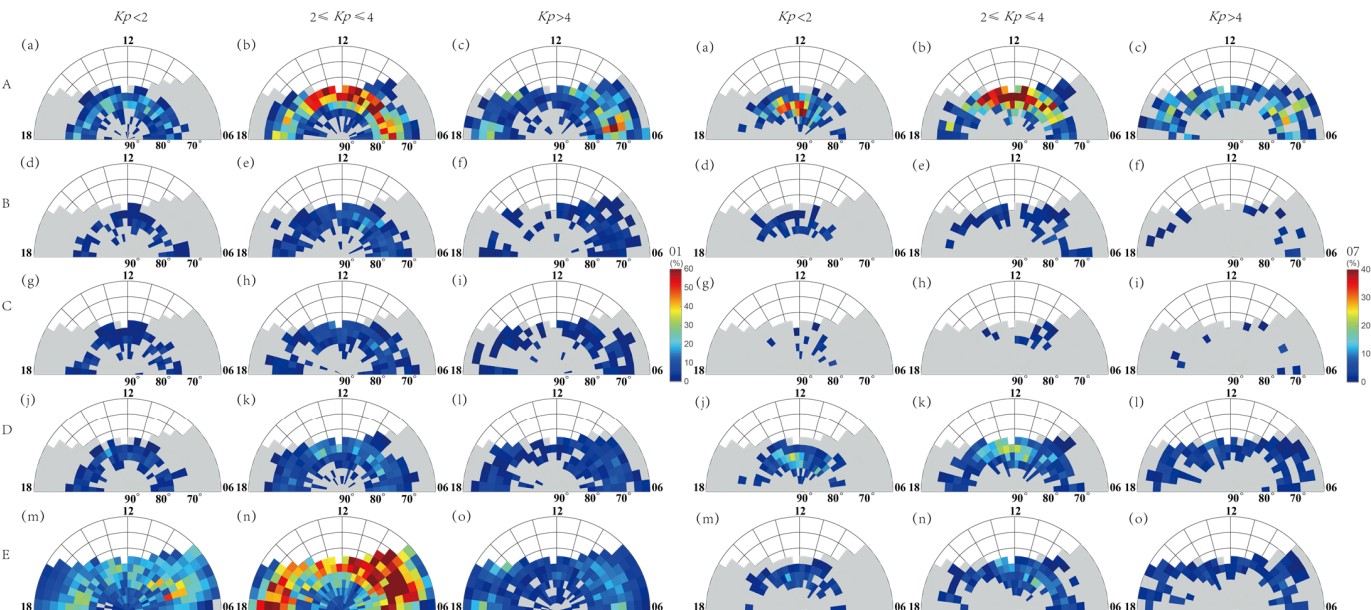

**Figure 4.** Temporal–spatial distribution of different types of ion upflows under different geomagnetic activities. From top to bottom are ion upflows of type A to E, respectively. The three columns on the left are statistics for January, and the right are statistics for July. (**a**,**d**,**g**,**j**,**m**) Quiet magnetic activities, (**b**,**e**,**h**,**k**,**n**) moderate magnetic activities, (**c**,**f**,**i**,**l**,**o**) disturbed magnetic activities.

During days with quiet magnetic activities, types A and D mainly occur at 0900–1500 MLT and 75–80° MLAT in prenoon and postnoon. Type B mainly occurs at 75–80° MLAT and 0600–1200 MLT in January, while 0900–1500 MLT in July. Type C mainly occurs at the range of 75–80° MLAT at dawn and dusk. Type E mainly occurs at MLAT range above 70° in dawn side of 0600–0900 MLT in January, while it mainly occurs at the range of 75–80° MLAT and 0900–1500 MLT in July. During days with moderate magnetic activities, type A and D are mainly distributed at 0900–1500 MLT and 70–80° MLAT. Type B is mainly distributed at 0600–1200 MLT and 70–80° MLAT. Type C is mainly distributed at 0600–1500 MLT and 70–80° MLAT. Type E mainly occurs at 0600–1200 MLT and 1500–1800 MLT, with MLAT range of 65–80° in January, while it mainly occurs at 0900–1500 MLT and 70–80° MLAT in July. During days with disturbed magnetic activities, type A and D mainly occur at MLAT range of 65–75° at dawn and dusk. Type B and C mainly occur at a MLAT range of 65–75° in the dawn side. Type E mainly occurs at the MLAT range of 65–80° in dawn and dusk in January, while it mainly occurs at the dawn side of 70–75° MLAT in July. This indicates that with the enhancement of geomagnetic activity, the main region of ion upflow expand to the lower latitude centered on the region in the quiet time. During days with moderate magnetic activities, the incidence of ion upflows is significantly enhanced.

### 3.2.3. The Effect of Interplanetary Magnetic Field (IMF) Components

Figures 5–7 shows the distribution of the average orbital incidences of various ion upflows under different directions of the component of the interplanetary magnetic field (IMF) (Bx, By, Bz).

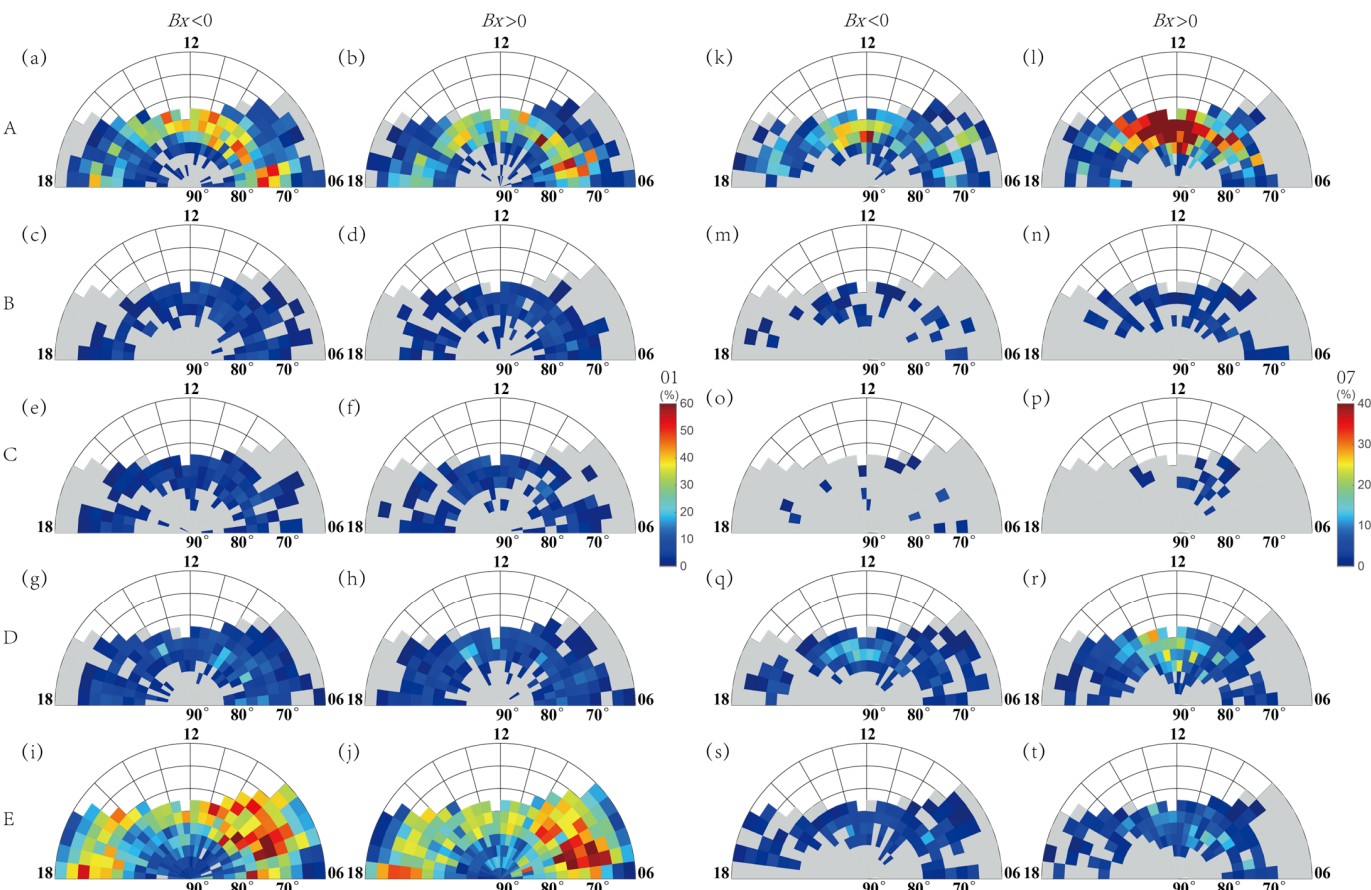

**Figure 5.** Temporal–spatial distribution of different types of ion upflows under different direction of IMF *Bx*. From top to bottom are ion upflows of type A to E, respectively. (**a–j**) statistics for January, (**k–t**) statistics for July.

Figure 5 shows the temporal–spatial distribution of ion upflows in different directions of the *Bx* component. The results show that when the direction of *Bx* changes, all kinds of ion upflows show different distribution characteristics at different latitudes and local time regions. The incidences of various ion upflows increase at MLT range of 60–70° as *Bx* < 0. During 70–80° MLAT, the incidences of type A and D increase in January and decrease in July, while the incidences of type B, C and E decrease in January and July. The incidences of various events decrease at a MLAT range above 80°. When *Bx* > 0, in the region of 0600–0900 MLT, the incidences of type A decrease in January and increase in July. The incidences of type B and E increase, and that of type C and D decrease. In the region of 0900–1200 MLT, the incidences of all kinds of ion upflows decrease in January and increase in July. In the region of 1200–1500 MLT, the incidences of type A decrease in January and increase in July, that of type B and C increase in January and decrease in July, and that of type D and E increase in January and July. In the region of 1500–1800 MLT, the incidences of all kinds of ion upflows increase.

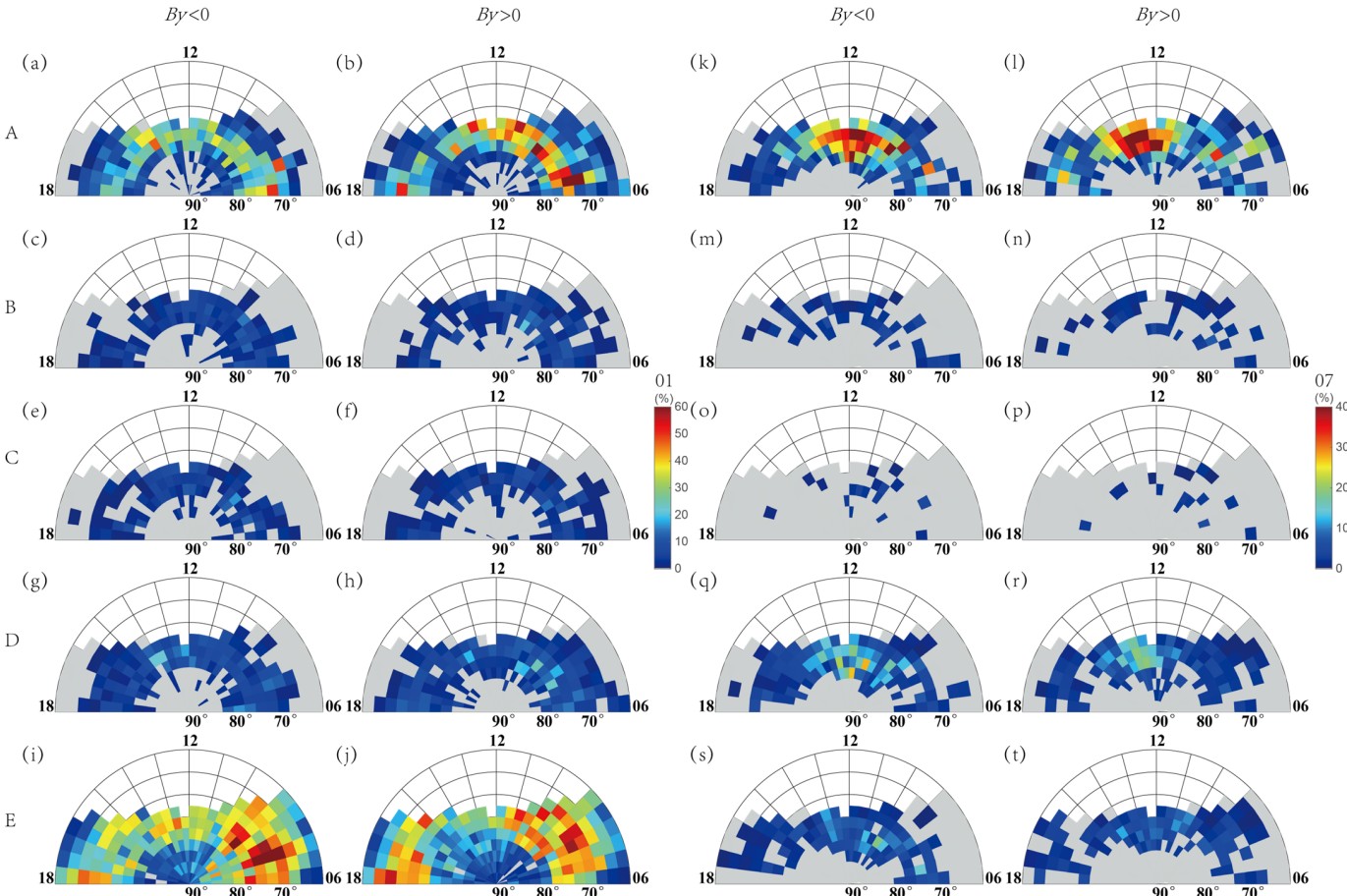

**Figure 6.** Temporal–spatial distribution of different types of ion upflows under different direction of IMF *By*. From top to bottom are ion upflows of type A to E, respectively. (**a**–**j**) statistics for January, (**k**–**t**) statistics for July.

As shown in Figure 6, the direction of the *By* component mainly affects the prenoon/postnoon incidences of type A and D, as well as type C in January and type E in July. When *By* < 0 (>0), the prenoon/postnoon incidences of type A in January and July are 4.82 (7.82)/5.01 (5.24) and 6.77 (3.80)/4.63 (6.59), respectively. The incidences of type C in January are 0.90 (0.93)/1.04 (0.69), respectively. The incidences of type D are 1.54 (2.10)/2.01 (1.24) in January and 2.65 (1.26)/2.18 (2.65) in July, respectively. The incidences of type E in July are 1.67 (0.98)/1.21 (1.10). This indicates that the direction of *By* may lead to the high-incidence area reversal in the prenoon or postnoon region.

Figure 7 shows the temporal–spatial distribution of ion upflows in different directions of *Bz* component. The results show that the direction of *Bz* affects both the region and intensity of ion upflows. When *Bz* < 0, the incidences of all kinds of ion upflows increase at an MLAT range of 60–75°, while they decrease at an MLAT range above 75°.

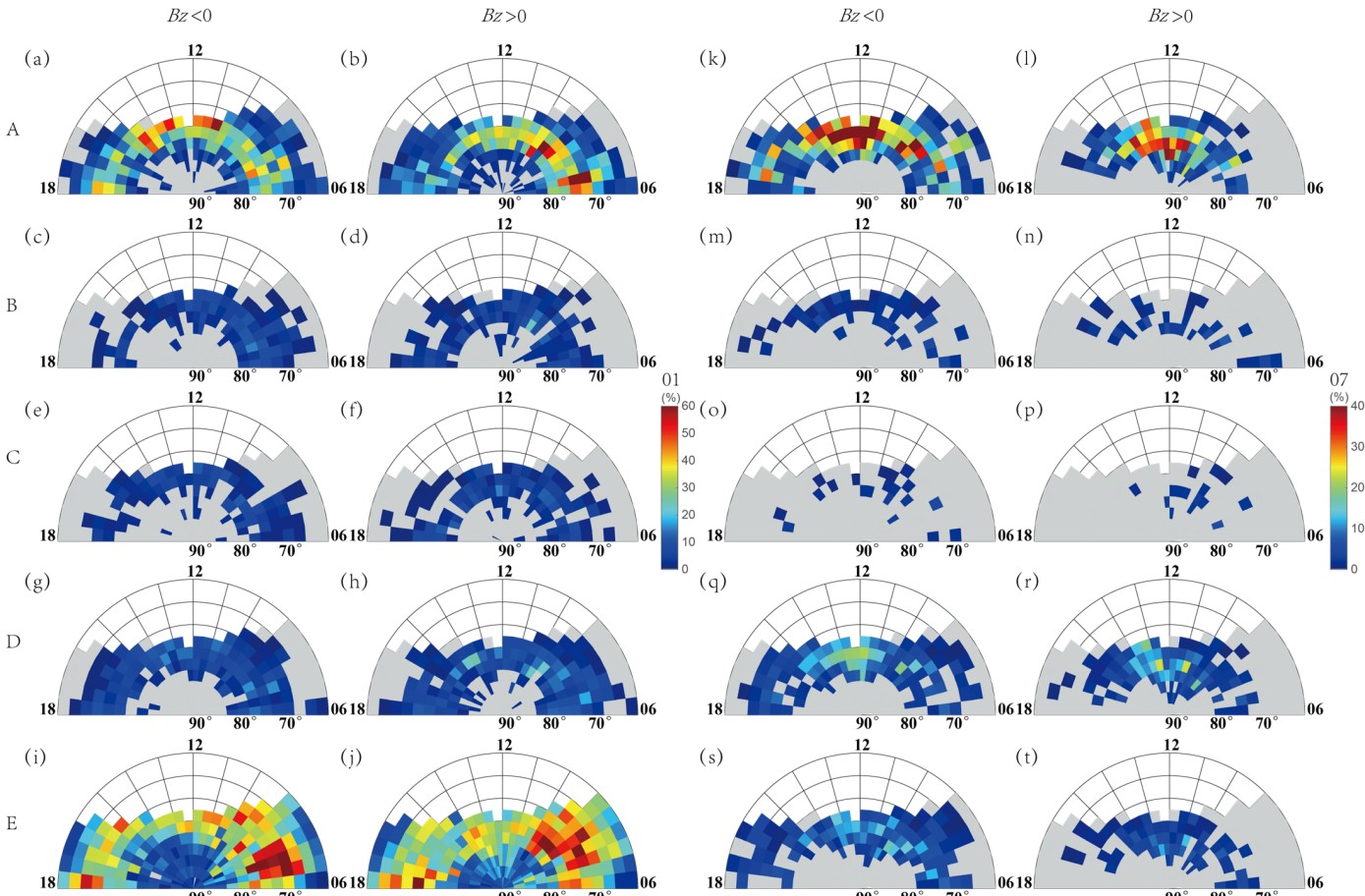

**Figure 7.** Temporal–spatial distribution of different types of ion upflows under different direction of IMF *Bz*. From top to bottom are ion upflows of type A to E, respectively. (**a–j**) statistics for January, (**k–t**) statistics for July.

### 3.3. The Distribution and Velocities for Different Types of Ion Upflows

Figure 8 shows the distribution of the average upflow velocity of all kinds of events on the AACGM coordinate plane (a–j is the statistical distribution of events; k–t is the temporal–spatial distribution of velocity), and the grid division remains unchanged. The statistical standard of the velocity is the peak velocity in the ion upflow and the region where it resides. Within a grid, the average upflow velocity is defined as the average of the peak velocity of each event. Each grid area must contain at least five ion upflows in order to ensure the validity of statistical results; otherwise, it will not participate in the statistics (the statistical data of type C in July and type B in Figure 9 do not comply with this standard, so there are no data in the velocity distribution diagram).

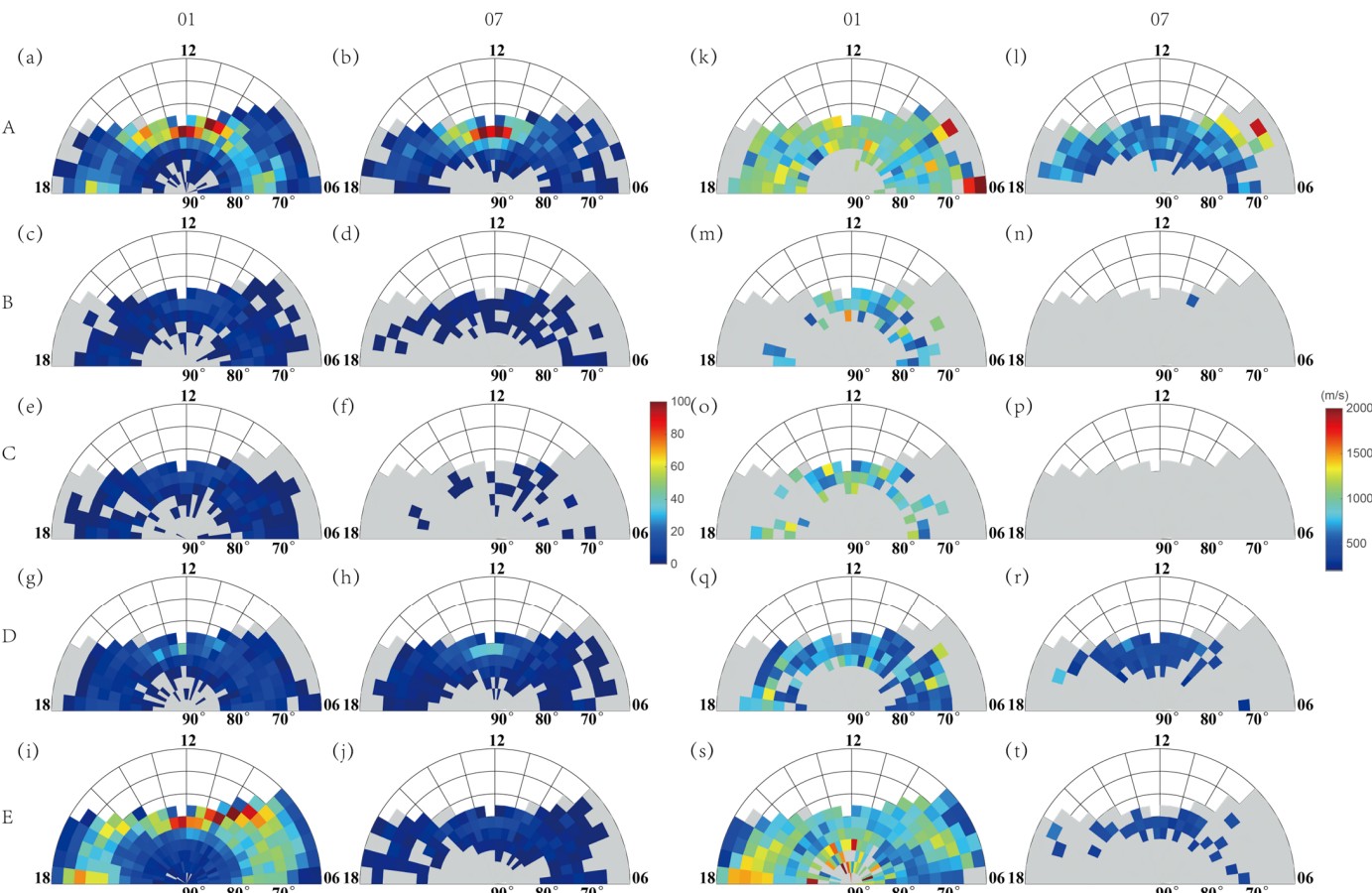

**Figure 8.** The distribution and velocities for different types of ion upflows in January/July in MLT/MLAT coordinate. From top to bottom are ion upflows of type A to E, respectively. (**a–j**) Total events for each month and type, (**k–t**) average velocities in different month and type.

As seen in Figure 8, type A has the highest upflow velocity, followed by type E, and type D has the lowest. The velocity in January is significantly higher than that in July. The upflow regions with higher velocities of all kinds of ion upflows are mainly concentrated at an MLAT region below 75°. The velocity is slightly higher in postnoon than in prenoon. Higher velocities of type A are in the region of 0600–0900 MLT, below 65° MLAT, and that of type E are in the region around 1700–1800 MLT and below 70° MLAT.

Figure 9 shows the velocity distribution of various ion upflows under different geomagnetic activity conditions. In the range of 70–80° MLAT, the upflow velocity of all kinds of events decrease with the increase in geomagnetic activity in January, while they increase with the increase in geomagnetic activity in July. The upflow velocity of type A is higher in the dusk side than in the dawn side during days with quiet and moderate geomagnetic activity, while it is lower in the dusk side than in the dawn side during days with disturbed magnetic activity. Under different geomagnetic activity conditions, the velocity of type E in dawn side is lower than that in the dusk side in January.

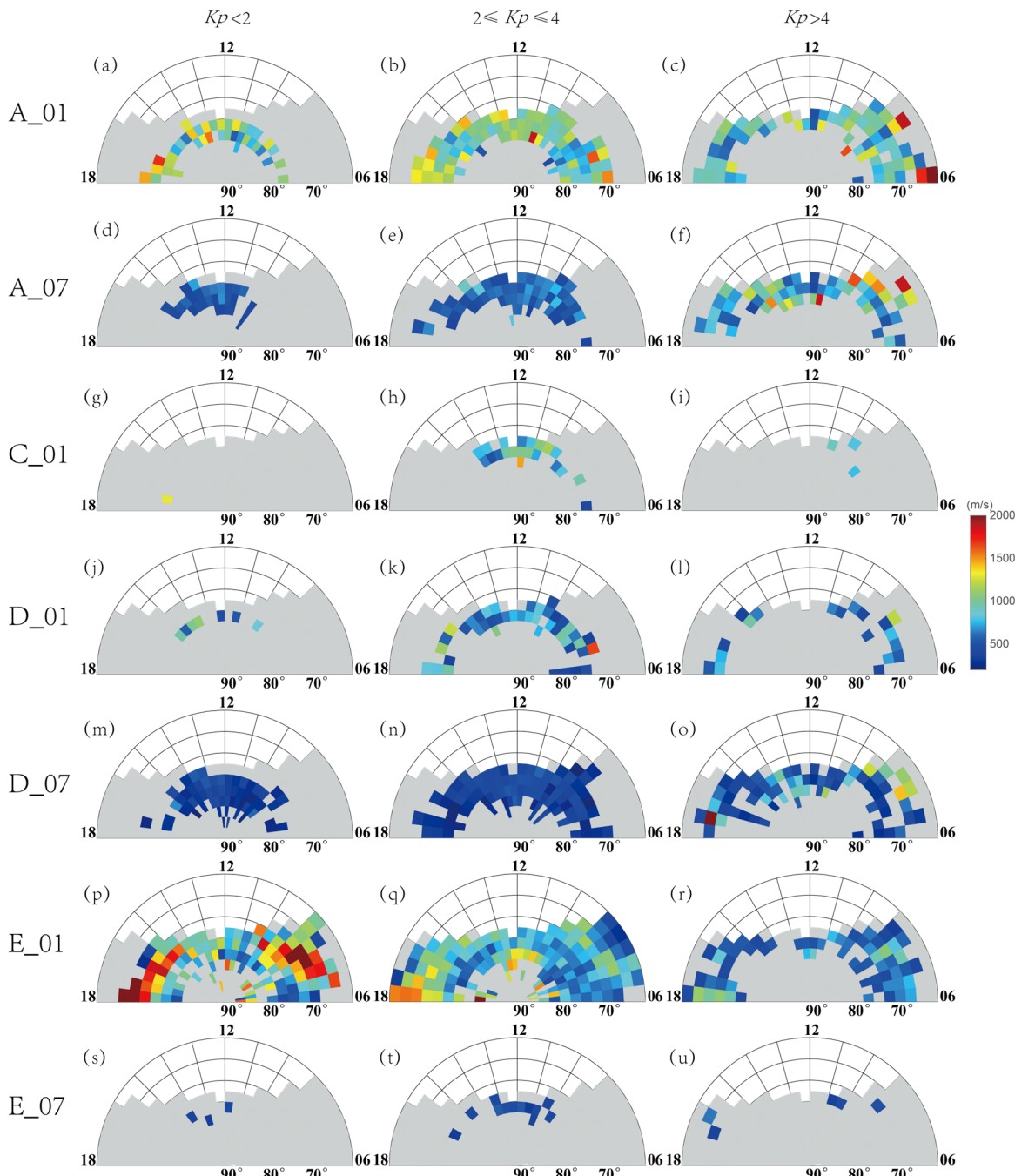

**Figure 9.** Temporal–spatial distribution of the average velocities of different types of ion upflows under different geomagnetic activities. (**a–f**) average velocities of type A in January/July, (**g–i**) average velocities of type C in January, (**j–o**) average velocities of type D in January/July, (**p–u**) average velocities of type E in January/July.

## 4. Discussion

Previous observations, based on the data of DE-2 satellite (Dynamics Explorer), have shown that the electron heating caused by soft electron precipitation at 800 km altitude can lead to ion upflows [27]. The soft electron precipitation results in ionization of neutral particles in the ionospheric F region and top ionosphere. The surrounding electrons are heated by coulomb collisions with the primary and secondary electrons, resulting in an upward bipolar electric field, through which the ions move upward [9,28]. The projection regions of Cusp and LLBL are the main regions of soft electron precipitation on a dayside

auroral oval. Observations of ESR (European Incoherent Scatter Svalbard radar) show that the ion upflows located in Cusp are mainly caused by soft electron precipitation [9]. The total energy of precipitating electrons near Cusp is about $1.6 \times 10^{-10}$ J m$^{-3}$, about 10 times of the total kinetic energy of upward ions near 350 km, so it can effectively accelerate the ionospheric upward flow [9]. For all kinds of ion upflows with electron acceleration characteristics, the number of type A and D in the region of Cusp and LLBL is not the largest relative to the total number of their respective types. However, for the Cusp region, almost only type A and D occur. For LLBL region, type A and D are also the two categories with the highest incidence. Therefore, type A and D in these two regions are mainly caused by the precipitation of soft electrons.

Kozlovsky et al. [29] made conjugate observations with the ultraviolet aurora imager of Polar satellite (UVI) and ESR radar. He found that in the postnoon aurora oval sector, at an altitude of 300–550 km, there are downward field-oriented ion flows towards the polar side of the aurora arc, and upward field-oriented ion flows on the equatorial side of the aurora arc. This upward/downward change in ion flow is thought to be caused by the difference in the vertical gradient of plasma density; that is, the convection associated with the aurora arc, the electron precipitation of the aurora arc, and the photoionization effect of sunlight together form the vertical gradient of plasma density near the aurora arc, which determines the bipolar diffusion and the observed ion motion. The relationship between the auroral arcs and ion upflows observed by Kozlovsky et al. [29] fits the definition of a type C ion upflow, in which the electron-accelerating structure (the auroral arc) exists only in part of the peak interval of the upflow velocity. In addition, type C occur mainly in the regions of LLBL and BPS, which are high-incidence areas of auroral arcs [13,30]. Therefore, ion upflows of type C may be related to the bipolar diffusion process formed by the vertical gradient of plasma density. Type B mainly occur in the regions of LLBL and BPS, similar to type C. However, unlike the single-electron acceleration structure of type C, type B has multiple electron acceleration structures. The LLBL and BPS regions are also high-occurrence areas of multiple auroral arcs [13,30]. Therefore, the multiple electron acceleration structures in type B correspond to multiple arcs, and the ion upflow is still related to the bipolar diffusion process.

CPS consists of two parts, one of which moves towards higher latitudes as the MLT increases on the dayside from dawn through noon, while the other connects to the former part and extends towards the nightside, where electrons enter the dayside from the nightside of plasma sheet. Electron precipitation in CPS is mainly "trapped" by a loss cone (forming mostly diffuse auroras), with hardly any precipitation caused by electron acceleration [23]. Due to the higher energy of the precipitating electrons, the deposition is caused to enter the ionosphere at a lower height, which mainly produces ionization to the atmosphere of ionospheric E region, while the heating of the initial height is weak. The statistical results show that the ion upflows related to soft electron precipitation and bipolar diffusion (type A–D) have low incidence in CPS, while type E has high incidence in CPS, indicating that type E ion upflows are caused by other formation mechanism, which required further study.

The polar cap region is a highly dynamic region. Due to the large energy flux from the magnetosphere, the ionosphere shows transient plasma flow and shear, and the electrons with higher energy precipitate to the lower height [31,32]. The upward movement of ions is attenuated by collisions with dense neutral particles. Thus, the acceleration of electrons in this region has less effect on the ion upflows. However, the high temperature of ions caused by friction heating in the polar cap region is closely related to the ion upflows, which can effectively drive ions upward [33].

Observations of the ESR radar show similar seasonal differences to those in Table 1 and Figure 2: the incidence is higher in winter than in summer, and the maximum occurs in the winter solstice and the minimum in the summer solstice [34–36]. Cohen et al. [37] investigated the effect of ionospheric density on ion upflows through a model simulation of ion upflow driven by auroral particle precipitation. The results indicate that as ionospheric

density increases, the incidences and velocities of ion upflows decrease, caused by decreases in electron heating and the bipolar electric field [23,37]. In summer, when the ionospheric high latitude is in sunshine, the extreme ultraviolet flux (EUV) increases, which leads to an increase in ionospheric density, resulting in less occurrence of ion upflows in summer than in winter.

The statistical results show that the incidence of ion upflows in Cusp on the dayside increases with $Kp < 3$, but decreases with $Kp > 3$ [38]. Figure 4 shows that with the enhancement of geomagnetic activity, the main upflow region of various events expands to the direction of low latitude with the region of quiet time as the center, and the peak incidence appears during days with moderate geomagnetic activity. This is because the enhancement of geomagnetic activity will lead to the enhancement of ionosphere–magnetosphere coupling, and the enhanced precipitation of high-energy particles and field-aligned electric potential difference can accelerate the upward movement of ions more effectively. The region of ion upflows moves towards the equator at low latitudes with the enhancement of geomagnetic activity [39], and the region of electron temperature enhancement also moves towards the equator with the increase in *AE* index [40]. During days with quiet geomagnetic activity, the auroral oval is located near 77° MLAT at midday on the dayside. As geomagnetic activity increases, the auroral oval expands towards equatorial and polar areas [41]. During days with moderate geomagnetic activity, the center of the upflow region is located near 75° MLAT, and the overlap of the auroral oval on the dayside increases [38,42], resulting in an increase in the observed particle precipitation and ion upflows. In addition, with the enhancement of geomagnetic activity, the intensity of ionospheric convection at low latitudes increases. The heating generated by the convection can further accelerate the ion upflows [24,39,43]. Thus, the incidence of ion upflows at low latitudes increased significantly.

The results in Figure 5 show that when the direction of *Bx* changes, the incidences in January and July show different characteristics. In the winter of the northern Hemisphere, the regions of 1300–1800 MLT and 0600–0900 MLT are the occurrence regions of hot spot aurora and auroral arcs with the magnetosphere source regions of BPS. When $Bx > 0$, the emission at 557.7 nm in this region is significantly enhanced [14,44], and the electron acceleration process of hot spot aurora and auroral arcs are also significantly enhanced. Similarly, the regions of 0900–1300 MLT are the occurrence regions of drapery dayside corona and radial dayside corona with the magnetosphere source regions of LLBL. The electron acceleration process of dayside corona is significantly enhanced when $Bx < 0$. Therefore, the incidences in the regions of 1300–1800 MLT (1500–1800 MLT for type A in January, and type B and C in July) and 0600–0900 MLT increase when $Bx > 0$, and that in the region of 0900–1300 MLT increase when $Bx < 0$. The possible reason for the differences between January and July is that the high-latitude ionosphere in summer is under sunshine conditions, and the solar extreme ultraviolet flux (EUV) increases, leading to the increase in ionospheric density, thus affecting the upward movement of ions. However, the relationship between EUV flux and IMF *Bx* is unknown.

Large-scale field-aligned currents are concentrated in two principal areas encircling the geomagnetic pole: region 1 is located near the poleward part of the field-aligned current region, and region 2 is located near the equatorward part. The shape of the auroral oval on the dayside is close to the field-aligned current in region 1 [45]. The current flows into the ionosphere on the dawn side and flows out on the dusk side. The prenoon (dawn) and postnoon (dusk) regions correspond to downward currents in prenoon and upward currents in postnoon, respectively. When IMF *By* is negative, an upward electric field is generated [46], superimposing the upward field-aligned current in the postnoon (downward field-aligned current in the prenoon) and enhancing (weakening) the net upward current in the region. At this point, the upward field-aligned current extends from the prenoon sector to the postnoon sector [47]. According to the ionospheric convection diagram, the convective vortex tilts towards the dusk side. When IMF *By* is positive, the convective vortex tilts towards the dawn side [48–51]. In Figure 6, the

inversion of the high-incidence area at the prenoon or postnoon region may be related to the ionospheric convection.

In the southward direction of IMF, the magnetic reconnection on the dayside is enhanced, and the latitude is lower than that in the northward direction. The enhanced precipitation of the boundary layer on the dayside leads to the enhancement of ionospheric ionization and accelerates the upward movement of ions in the ionosphere [52]. However, it can be seen from Figure 7 that the incidence in the low-latitude region increases significantly when IMF is southward, but the incidence in MLAT region above 80° is greater when IMF is northward. The possible reason is that the increase in the upflow velocity in the high-latitude region does not exceed 200 m/s when IMF is southward.

## 5. Conclusions

Based on the observations of particle precipitation and ion drift from the DMSP F13 in January and July 2005, the ionospheric ion upflows in dayside auroral oval can be divided into five types according to the velocity of ion upflows and the spectrum characteristics of auroral particle precipitation: (1) Type A: The upflow velocity has obvious peak interval, and the whole peak interval should have obvious single and continuous electron acceleration structure; (2) Type B: The upflow velocity has an obvious peak interval, corresponding to multiple and scattered electron acceleration structure; (3) Type C: The upflow velocity has an obvious peak interval, but only the ascending or descending segment corresponds to the structure of electron acceleration; (4) Type D: the upflow velocity corresponds to the structure of electron acceleration, but the velocity has no obvious change trend or complete peak interval; (5) Type E: the upflow velocity does not correspond to the structure of electron acceleration. The distribution characteristics of all kinds of ion upflows are analyzed statistically at the same time. The results show that, with the changes of geomagnetic activity, interplanetary magnetic field and seasons, there are significant differences in the distribution characteristics for different types of ion upflows. The results are summarized as follows:

(1)　The incidence of ion upflows in winter is higher than that in summer. Type A–D have the highest occurrence at MLAT range of 70–80°, which is 3–6 times of the total occurrence of other latitude ranges, and mainly appear in dayside regions of BPS, LLBL, Cusp and mantle. Type E have high incidence at MLAT range above 65°, and mainly appear in dayside regions of CPS, BPS and LLBL. The region of Cusp mainly contains type A and D. Type B and C mainly appear in LLBL and BPS. The region of CPS mainly contains type E. In January, all kinds of ion upflows mainly occur on the dawn and dusk side, and the incidence on the dawn side is higher than that on the dusk side, showing obvious "dawn–dusk asymmetry". While in July, all kinds of ion upflows mainly occur around magnetic noon, with a symmetric distribution centered at the magnetic noon.

(2)　With the enhancement of geomagnetic activity, the main upflow region of all kinds of events expand to the lower latitude centered on the region of the quiet geomagnetic activity. During days with moderate geomagnetic activity, the incidence increases significantly. When $Bx < 0$, the incidence increases significantly at MLAT region below 70°, as well as the regions of 0600–0900 MLT and 1500–1800 MLT. When the direction of $By$ changes, the occurrence of all kinds of ion upflows shows obvious high-incidence area reverse at the prenoon or postnoon region. When $Bz < 0$, the incidence increases significantly at MLAT region below 75°.

(3)　Type A ion upflow has the highest velocity of ion upflows, then is type E, and type D is the lowest. The average velocity of ion upflows in winter is significantly higher than that in summer. At MLAT range of 70–80°, the velocity of all kinds of ion upflows decrease with the increase of geomagnetic activity in January, while increase with the increase of geomagnetic activity in July.

**Author Contributions:** Conceptualization, Y.Y. and Z.-J.H.; methodology, Y.Y., H.-T.C. and Z.-J.H.; software, Y.Y. and Z.-J.H.; validation, Y.Y., H.-T.C. and Z.-J.H.; formal analysis, Y.Y.; investigation, Z.-J.H.; resources, Z.-J.H.; data curation, Y.Y.; writing—original draft preparation, Y.Y.; writing—review and editing, Y.Y.; visualization, Y.Y.; supervision, H.-T.C. and Z.-J.H.; project administration, Z.-J.H. and Y.-S.Z.; funding acquisition, Z.-J.H. and Y.-S.Z. All authors have read and agreed to the published version of the manuscript.

**Funding:** This work was supported by the National Natural Science Foundation of China (Grant Nos. 41831072, 41874195, 42130210), the National Key R&D Program of China (Grant Nos. 2021YFE0106400, 2018YFC1407303), Space Science Pilot Project of the Chinese Academy of Sciences (grant number XDA15350202), a fund from Institute of Applied Meteorology, and the Shanghai Pujiang Program.

**Data Availability Statement:** DMSP SSJ/SSIES data were provided by The Johns Hopkins University Applied Physics Laboratory (http://sd-www.jhuapl.edu/Aurora/, accessed on 11 December 2020.) and DMSP Space Environment Data Access (https://www.ngdc.noaa.gov/stp/satellite/dmsp/index.html, accessed on 9 December 2020.). Interplanetary magnetic field data were from NASA/GSFC OMNI website (http://omniweb.gsfc.nasa.gov), accessed on 5 March 2021. *Kp* index was provided by the German earth science Research center (GFZ), Data sources, (ftp://datapub.gfz-potsdam.de/download/10.5880.Kp.0001), accessed on 14 January 2022. IGRF geomagnetic field data are available from website http://www.geomag.bgs.ac.uk/research/modelling/IGRF.html, accessed on 19 August 2022.

**Conflicts of Interest:** The authors declare no conflict of interest.

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
