# Peer review of "Classification and Distribution of the Dayside Ion Upflows Associated with Auroral Particle Precipitation"

_universe, doi:10.3390/universe9040164_

Round 1

Reviewer 1 Report

This paper proposed a statistical study of different types ion upflows involving DMSP 13 observations during January and July 2005. The results showed significant differences in the characteristics of the five types of ion upflows. This statistical study could contribute to the understand of the relationship between the particle precipitation and ion upflows. This reviewer considers the following revisions before its publication.

1. Why do you choose the observations from DMSP 13 and the time period of January and July 2005?

2. “A total of 15,198 ion upflow events occur in 178 the range of 60~90° MLAT in the Northern Hemisphere during January and July 2005 are 179 statistically studied.”. How did you identify these events? Manually or programmatically?

3. Line 160: the format of time text “08:20:41” is better.

4. Fonts in Figures 1, 4, 5 are too small to see the text. Can you resize the fonts or re-layout the figure?

Reviewer 2 Report

The manuscript discusses the classification and distribution of dayside ion upflows linked to auroral particle precipitation, using data from the DMSP F13 satellite. Based on my assessment, it falls within the scope of the journal and is well-written, interesting, and presents findings that merit publication. I have included some suggestions, corrections, and other considerations below.

Line 13: ...the DMSP F13 satellite in January and...

Line 32:... Type A events have .... --> Type A ion upflow has ... [Along the text upflows and events are used as synonymous. Be carefull to well contextualize the meaning and their correct linguistic use]

Line 74:... ...78° ILAT --> ...78° MLAT [Was it meant MLAT? please, check it]

Line 99: [Few additional info about the satellite can be appreciated by the readers,  as when it was launched, by who and for what porposes]

Line 103: ... the digital and ... [I do not understand the term digital in this context]

Line 138: [It would be useful here to mention how the data from the IGRF-11 model are used to derive the values of Vb ]

Line 178: ...satellite, a total of ....

Line 185:... , Cusp and plasma mantle...

Line 199:... (CPS), while the others are events with...

Line 205:.. The AACGM coordinate [The first time an acronym is used needs to be written with the extended names]

Line 234 FIGURE 3 [Numbers 18, 12 and 06 together with latitudes need to be magnified. Such consideration is valid also for the following figures]

From line 269 to 282 [The noun INCIDENCE is singular so the verb should agree, please check the grammar in those sentences]

Line 328:.. during days with quiet and moderate...

Line 329:.. during days with disturbed...

Line 331:.. side was lower than that ... [Unclear, maybe a conjunction is missing]

Line 336:... of DE-2 observation [DE-2 (??) never introduced before in the text]

Line 343:... ESR radar [ESR (??) never introduced before in the text]

Line 417:... expands towards equator and polar areas ...

Line 443:...region 1 [Please, recall briefly what is region 1 for readers benefit]

Line 509:...field data are available from ....
